# Examining doctors' business analytics capabilities in using the electronic medical record system for decision-making effectiveness in intensive care units: Impact of the COVID-19 pandemic

**Ewilly Jie Ying Liew**[1]*, **Andrei O. J. Kwok**[2], **Sharon G. M. Koh**[3], **Shairil R. Ruslan**[4], **M. Shahnaz Hasan**[4], **Yeh Han Poh**[5]

**1** Department of Econometrics & Business Statistics, School of Business, Monash University Malaysia, Subang Jaya, Malaysia, **2** Department of Management, School of Business, Monash University Malaysia, Subang Jaya, Malaysia, **3** Department of Economics, School of Business, Monash University Malaysia, Subang Jaya, Malaysia, **4** Department of Anaesthesiology, Faculty of Medicine, Universiti Malaya, Kuala Lumpur, Malaysia, **5** Department of Anaesthesiology and Intensive Care, Hospital Kuala Lumpur, Kuala Lumpur, Malaysia

* ewilly.liew@monash.edu

## Abstract

### Background

Advancements in electronic medical record (EMR) systems raise the demand for doctors' digital and analytical skills to process large-scale healthcare data for evidence-based decisions. The present challenge arises with the need to understand how doctors can develop business analytics capabilities using the EMR system for decision-making from an end user's perspective.

### Aim

Integrating the technology acceptance model and the business analytics model for healthcare, this study examines how individual doctors' technology perceptions of using an EMR system influence their ability to develop business analytics capabilities for making effective healthcare decisions in intensive care units (ICUs). The research questions are: How do doctors' perceptions of using an EMR system influence their ability to develop business analytics capabilities? and How do doctors' business analytics capabilities affect the effectiveness of their healthcare decisions? This study focuses on the context of using the EMR system as a business analytics-enabled architecture rather than a general information system.

**Data availability statement:** All relevant data from this study are publicly available from the figshare repository (https://doi.org/10.6084/m9.figshare.23723313.v1).

**Funding:** This study was funded by two internal grants from Monash University Malaysia. (1) Global Asia in the 21st Century (GA21) Multidisciplinary Funding Platform under the project title, "Leveraging dynamic capabilities for healthcare analytics through quality use of technology" (grant number: GA-MA-16-L01, received by authors EL, SK, YHP, and SH). (2) COVID Kickstarter Fund Scheme under the project title, "Enhancing electronic medical record (EMR) use behaviors for healthcare analytics: a dynamic capabilities perspective" (grant number: Covid-2-2020, received by authors EL, SK, AK, and YHP). The funders had no role in study design, data collection and analysis, decision to publish, or preparation of the manuscript. There was no additional external funding received for this study.

**Competing interests:** The authors have declared that no competing interests exist.

## Methods

We surveyed a final sample of 130 ICU doctors from public tertiary hospitals in Malaysia, a developing country. This study uses PLS-SEM to analyze two phases, comparing doctors' technology perception and business analytics capabilities before and during the pandemic.

## Results

We found significant shifts in ICU doctors' perceptions of using the EMR system (i.e., perceived ease of use and usefulness) influencing the development of their business analytics capabilities (i.e., data aggregation, data analysis, and data interpretation) for decision-making effectiveness. Data analysis was the only capability contributing to decision-making effectiveness during the pandemic. Significant differences in the relationships were observed before and during the COVID-19 pandemic.

## Conclusion

We demonstrate that COVID-19 has accelerated favorable technology perceptions and the increasing dependency on developing business analytics capabilities to inform healthcare decisions. Our findings contribute to the critical importance, challenges, and opportunities of using the EMR system for more data-driven decision-making, especially in the post-COVID era.

## Introduction

Investment in health information systems grows with heightening awareness and technological advancement in the healthcare industry [1,2]. Adopting an electronic medical record (EMR) system enables hospital data standardization and practice. More than a data repository, public tertiary hospitals can capitalize on EMR systems to efficiently manage large volumes of medical and patient records for informing evidence-based decisions [3,4].

In the care and management of critically ill patients, the EMR system becomes a vital support tool aiding doctors in treatment plans [5]. With increased accessibility to valuable information all in one place, healthcare professionals can make informed decisions on accurate diagnosis, treatment, and medication to improve patient care [6]. The ability to access information, the speed of making decisions, and the correct understanding of patients' diagnoses contribute to decision-making effectiveness (DME) [7]. In this study, we define DME as the extent to which public tertiary hospitals could achieve the objective of serving the general public's healthcare needs within time and resource constraints. More so in the setting of the intensive care unit (ICU), where critically ill are cared for, decisions that affect mortality and morbidity outcomes are higher compared to the general wards which house relatively stable patients [8].

Extant information systems (IS) studies tend to understand the EMR system's analytics capabilities in terms of the performance ability of its business analytics

functionalities [9]. We believe the success of an EMR system implementation depends on whether end users can recognize the benefits of using various system functionalities for decision-making [5,10,11]. The health IS literature further relates the EMR system to individual healthcare professionals, proposing that their technology perceptions as primary end users can influence their ability to use the EMR system for patient care [11]. As system users, doctors should develop appropriate digital and analytical skills to derive these recognized benefits from large volumes of EMR data and inform evidence-based decisions [12–14]. Thus, DME can be achieved along the business objectives established by the hospital management [15].

The business analytics (BA) literature remains scarce about the efficacy of using the EMR system's functionalities to improve individuals' BA capabilities for decision-making [16,17]. Past studies tend to lean toward the cost and architectural design effectiveness of an EMR system. Only a few studies have examined the end users' perspective of using an EMR system to make effective healthcare decisions [18]. Hence, we propose to approach the EMR system as a BA-enabled architecture that supports individual users making informed decisions rather than merely a general IS architecture.

In this study, we recognized the importance of human factors behind using an EMR system. The problem arises when doctors themselves are ambiguous about the benefits of using the EMR system and lack the analytical skills to derive these benefits from using the EMR system for healthcare decisions [19]. Therefore, addressing these problems would require a closer examination of the interrelationships between perceptions and benefits of using an EMR system from the end users' perspective. We raise the following research questions:

*RQ1: How do doctors' perceptions of using an EMR system influence their ability to develop business analytics capabilities?*

*RQ2: How do doctors' business analytics capabilities affect the effectiveness of their healthcare decisions?*

## Conceptual model

We integrate Davis' [20] technology acceptance model (TAM) and Wang and Byrd's [7] business analytics model in healthcare. Our conceptual model aims to examine how ICU doctors' perceptions of using an EMR system would influence their development of business analytics (BA) capabilities for decision-making effectiveness (DME) in public tertiary hospitals [Fig 1].

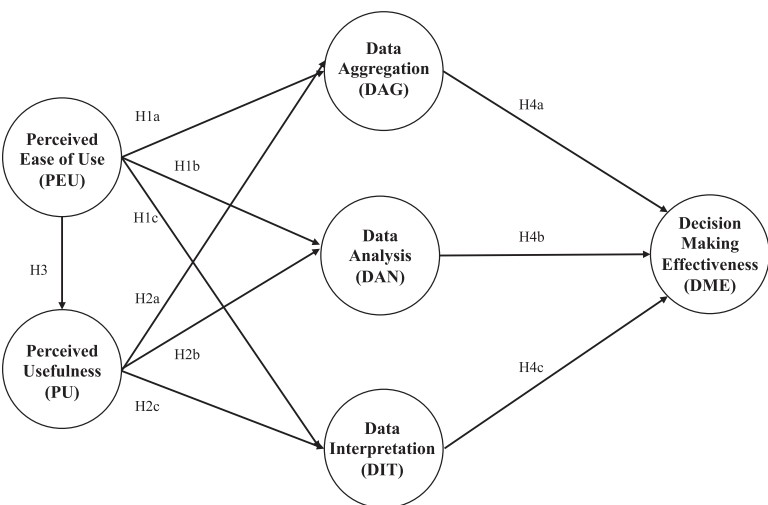

**Fig 1. Conceptual model.**

The TAM has been widely used to explain individuals' IS adoption and usage behavior [21,22] based on two core predictors: perceived ease of use (PEU) and perceived usefulness (PU). We propose that doctors are more likely to use the EMR system to develop their BA capabilities when it is easy to use and useful for performing their daily healthcare operations. Higher PEU and PU offer convenience and utility for ICU doctors already busy with critical patient care [23]. These benefits imply lower cognitive, emotional, and physical stress when using the EMR system [24].

We also propose that PEU is a precondition that should directly influence PU but not vice versa [25]. ICU doctors who believe that the EMR system is accessible and easy to use are more likely to find it useful for developing their BA capabilities.

Wang and Byrd [7] postulated the system's BA capabilities in three dimensions: data aggregation (DAG), data analysis (DAN), and data interpretation (DIT). In contrast, we focus on using the EMR system for its BA functionalities from the end users' perspective. We believe ICU doctors can use the EMR system as a tool to develop their BA capabilities in terms of the individuals' ability to perform DAG, DAN, and DIT for DME.

In the IS literature, decision-making effectiveness (DME) is a critical outcome variable [26] to measure system performance. Literature has evidenced that real-time clinical and patient data recording augments doctors' DME in diagnostic accuracy and clinical knowledge about patient illnesses [9,27]. The EMR system integrates healthcare databases across the hospital, enabling doctors to access and process the integrated healthcare information on a unified system interface in real-time [28]. Therefore, doctors could synthesize and analyze data from various sources to gain a multidimensional perspective on a patient or an illness that needs timely and accurate treatment prescriptions.

Doctors need data analysis skills to perform descriptive, predictive, and prescriptive analytics on patient care in the ICU. These skills involve identifying common patterns of patient data (descriptive analytics), building predictive models of acute illnesses (predictive analytics), and interpreting the analysis results accurately to prescribe a suitable treatment plan (prescriptive analytics) for patients under close monitoring [29]. The business analytics functionalities of an EMR system require doctors to be able to use the system well for its desired functionalities. Therefore, doctors' digital capabilities of using the business analytics functionalities of an EMR system are vital to decision-making effectiveness. The effectiveness of doctors' healthcare decisions should be commensurate with the increased speed and improved quality of decisions to optimize resource allocation and ensure quality care [30].

Studies are limited about how the pandemic has shifted how doctors use an EMR system for improved decision-making and patient care. At COVID-19 referral hospitals, doctors face pressure to use the EMR system to quickly synthesize patient health history, analyze pathological symptoms, and interpret clinical diagnoses in prescribing the correct treatment while operating at overcapacity. Hence, we would extend our earlier studies [31,32] to compare the changes in doctors' perceptions of using the EMR system and their development of BA capabilities before and during the pandemic.

### Hypotheses

We analyzed the following four main hypotheses:

H1: PEU directly and positively influences doctors' use of the EMR system for developing BA capabilities in terms of DAG (H1a), DAN (H1b), and DIT (H1c).

H2: PU directly and positively influences doctors' use of the EMR system for developing BA capabilities in terms of DAG (H2a), DAN (H2b), and DIT (H2c).

H3: PEU directly and positively influences PU of using the EMR system.

H4: Using the EMR system for developing BA capabilities in terms of DAG (H4a), DAN (H4b), and DIT (H4c) directly and positively influences doctors' DME.

## Methods

### Data collection and sampling design

The first round of data collection (Time 1) was administered in mid-2018 before the pandemic, followed by the second round of data collection (Time 2) in mid-2020 during the pandemic. The Time 1 survey was distributed using paper-based and online questionnaires as part of a more extensive study conducted over two years in the ICU of a public tertiary hospital that had implemented an in-house developed EMR system. The Time 2 survey was extended using online questionnaire only during the pandemic to include the ICUs of another four public tertiary hospitals. All five hospitals were the referral centers directly involved in the intensive care management of COVID-19-related patients.

We employed the cluster random sampling method to survey ICU doctors currently attending to critical care in a public tertiary hospital and using the EMR system for decision-making. Doctors refer to individuals with MD degrees and are the primary decision-makers about patient care and diagnosis [25]. Five ICUs were randomly selected from a list of public tertiary hospitals in Malaysia's capital city. The survey invitation was sent out to all doctors involved in critical care at the selected ICUs. Doctors were recruited using questionnaires over two distinct periods. We could only recruit ICU doctors who were present during our visit to the hospital before the pandemic (Time 1) and through snowballing contacts among the closely knitted ICU doctor groups in the hospital during the pandemic (Time 2).

Participant consent was implied upon returning the completed questionnaire. Participants may withdraw at any stage or avoid answering questions that are felt too personal or intrusive. The anonymity of doctors' participation was ensured, and only the combined results of all participants were analyzed. For this study, our data collection procedure complied with the ethical approval (NMRR-21-277-58120 (IIR)) obtained from the Medical Research and Ethics Committee (MREC), Ministry of Health Malaysia (MOH).

### Survey instruments

We designed the questionnaire using established instruments from past information systems (IS) and business analytics (BA) literature to ensure content validity.

Items measuring PU and PEU were adapted from Sykes, Venkatesh [25] to measure doctors' technology perceptions of using the EMR system. PU refers to *"the degree to which a person believes that using a particular system would enhance his or her job performance"* [20] while PEU refers to *"the degree to which a person believes that using a particular system would be free from effort"* [20].

Items measuring DAG, DAN, DIT, and DME were modified from Wang and Byrd [7]. The questions were modified to ask how the EMR system can effectively enable doctors to perform DAG, DAN, and DIT, focusing on doctors as the subject matter. DAG, DAN, and DIT represent the three dimensions of doctors' business analytics capabilities when using the EMR system. DAG involves data generation and pre-processing, which includes transforming different types of healthcare data into data format readily readable by any analysis platform. DAN performs appropriate analyses to transform raw data into meaningful information that informs evidence-based decisions. DAN includes descriptive analytics (summary statistics), predictive analytics (predicting the future based on historical data), and prescriptive analytics (suggesting what will happen in the future). DIT produces reports and chart summaries that provide a comprehensive view of patients' conditions and warnings for clinical surveillance. The outcome variable of our study, DME refers to *"the extent to which a decision achieves the objectives established by management at the time it is made"* [15]. Please refer to the S1 Table for a complete list of survey instruments used in this study.

### Statistical analysis

Our multivariate analysis employed the partial least square structural equation modeling (PLS-SEM) technique using SmartPLS 3 software [33]. PLS-SEM is a variance-based method [34] that does not require normality distributional

assumption to analyze complex models with smaller samples [35]. The prediction-oriented nature of PLS-SEM is appropriate for exploring the theoretical integration of IS and BA constructs and analyzing the interrelationship between these constructs that can explain doctors' decision-making effectiveness (DME) in the ICU. The method also works well for testing multiple hypotheses from a relatively homogenous population [36], such as the ICU doctor population in this study.

According to Hair, Risher [35] and Benitez, Henseler [37] guidelines, we performed the PLS-SEM analysis in the following three stages: [i] measurement model assessment, [ii] between-sample comparison based on the measurement invariance of the composite models (MICOM) procedure and the multi-group analysis (MGA), and [iii] structural model assessment.

We also assessed common method bias based on a latent common method factor approach [38] using PLS-SEM [39], to mitigate the risk of self-reporting bias commonly found in questionnaire survey [39]. We found insignificant and small magnitudes of method variance, indicating that common method bias was unlikely a severe concern for this study. Please refer to the S2 Table for the assessment result.

## Results

The final sample consisted of 130 valid responses, combining Time 1 responses (n1 = 67) before and Time 2 responses (n2 = 63) during the COVID-19 pandemic. Comparing Time 1 and Time 2 periods indicates a shift in doctors' technology perceptions and their business analytics capabilities development, given the adjustment and time learning to use the EMR system. The Time 2 survey also coincides with the pandemic, significantly accelerating doctors' reliance on the EMR system. However, the representativeness test revealed no significant differences in sample characteristics (age, gender, ethnicity, preferred language, position, and experience) between Time 1 and Time 2 samples at the 1% significance level. Table 1 summarizes the final sample profile.

**Table 1. Sample Profile.**

| Variables | Subgroups | Time 1[a] | Time 2[b] | Total | p |
|---|---|---|---|---|---|
| **Age (year)** | | 33.448 ± 3.081 | 32.222 ± 3.108 | | 0.026** |
| **Gender** | Male | 33 [25.4%] | 25 [19.2%] | 58 [44.6%] | 0.357 |
| | Female | 34 [26.2%] | 38 [29.2%] | 72 [55.4%] | |
| **Ethnicity** | Malay | 20 [15.4%] | 23 [17.7%] | 43 [33.1%] | 0.881 |
| | Chinese | 34 [26.2%] | 29 [22.3%] | 63 [48.5%] | |
| | Indian | 12 [9.2%] | 10 [7.7%] | 22 [16.9%] | |
| | Others | 1 [0.8%] | 1 [0.8%] | 2 [1.5%] | |
| **Preferred language** | English | 62 [47.7%] | 53 [40.8%] | 115 [88.5%] | 0.345 |
| | Bahasa Malaysia | 4 [3.1%] | 6 [4.6%] | 10 [7.7%] | |
| | Mandarin Chinese | 1 [0.8%] | 2 [1.5%] | 3 [2.3%] | |
| | Tamil | 0 [0.0%] | 2 [1.5%] | 2 [1.5%] | |
| **Position** | Specialist | 10 [7.7%] | 9 [6.9%] | 19 [14.6%] | 0.993 |
| | Medical Officer | 55 [42.3%] | 52 [40.0%] | 107 [82.3%] | |
| | House Officer | 2 [1.5%] | 2 [1.5%] | 4 [3.1%] | |
| **Experience** | < 5 years | 41 [31.5%] | 28 [21.5%] | 69 [53.1%] | 0.037** |
| | 6 - 10 years | 25 [19.2%] | 28 [21.5%] | 53 [40.8%] | |
| | 11-15 years | 0 [0.0%] | 6 [4.6%] | 6 [4.6%] | |
| | > 15 years | 1 [0.8%] | 1 [0.8%] | 2 [1.5%] | |

Note: ** p < 0.05.

Continuous values: mean ± standard deviation. Categorical values: frequency, n (%)

[a] Time 1: before COVID-19 pandemic. [b] Time 2: during COVID-19 pandemic.

## Measurement model assessment

The measurement model satisfied all indicator and internal consistency reliability, convergent validity, and discriminant validity criteria. Table 2 shows that all item loadings were above 0.708 and significant at the 1% significance level with sufficient indicator reliability. Only one item (PU04) had an indicator loading of 0.659, below the recommended threshold of 0.708. Consequently, the item PU04 was examined and retained in the model because its latent PU construct still met the construct reliability and validity criteria [37].

Based on Table 2, the constructs' internal consistency reliability was supported with all Dijkstra-Henseler's $\rho_A$ greater than 0.708 [35]. Dijkstra-Henseler's $\rho_A$ lies between the unweighted Cronbach's alpha [conservative] and the weighted composite reliability [liberal] to approximate exact composite reliability. All average variance extracted (AVE) was greater than 0.50, evidencing convergent validity where all latent constructs explained more than 50% of the variance in their observed items [35].

Discriminant validity between constructs was assessed based on Henseler, Ringle's [40] heterotrait-monotrait [HTMT] ratio of correlations. Table 3 shows that most HTMTs were lower than the $HTMT_{.85}$ criterion [41], one was lower than the $HTMT_{.90}$ criterion, and two were above the $HTMT_{.90}$ criterion [42]. Borderline discriminant validity appeared among the BA constructs (DAG, DAN, and DIT) and the IS constructs (PU and PEU). These constructs shared considerable conceptual

**Table 2. Indicator Reliability, Internal Consistency Reliability and Convergent Validity.**

| Constructs | Item | Loadings | $\rho_A$ | AVE |
|---|---|---|---|---|
| **Perceived Usefulness (PU)** | PU01 | 0.888*** | 0.884 | 0.746 |
|  | PU02 | 0.934*** |  |  |
|  | PU03 | 0.941*** |  |  |
|  | PU04 | 0.659*** |  |  |
| **Perceived Ease of Use (PEU)** | PEU01 | 0.842*** | 0.921 | 0.803 |
|  | PEU02 | 0.918*** |  |  |
|  | PEU03 | 0.910*** |  |  |
|  | PEU04 | 0.913*** |  |  |
| **Data Aggregation (DAG)** | DAG01 | 0.902*** | 0.929 | 0.875 |
|  | DAG02 | 0.956*** |  |  |
|  | DAG03 | 0.948*** |  |  |
| **Data Analysis (DAN)** | DAN01 | 0.935*** | 0.927 | 0.814 |
|  | DAN02 | 0.913*** |  |  |
|  | DAN03 | 0.898*** |  |  |
|  | DAN04 | 0.861*** |  |  |
| **Data Interpretation (DIT)** | DIT01 | 0.936*** | 0.931 | 0.876 |
|  | DIT02 | 0.940*** |  |  |
|  | DIT03 | 0.932*** |  |  |
| **Decision-Making Effectiveness (DME)** | DME01 | 0.923*** | 0.928 | 0.872 |
|  | DME02 | 0.955*** |  |  |
|  | DME03 | 0.923*** |  |  |

Note:

*** p<0.001,

** p<0.01,

* p<0.05 based on bootstrapped 5000 subsamples on one-tailed test

**Table 3. Discriminant Validity.**

| Constructs | DAG | DAN | DIT | DME | PEU |
|---|---|---|---|---|---|
| **DAN** | 0.895 [0.838,0.940] | | | | |
| **DIT** | 0.811 [0.709, 0.883] | 0.951 [0.909, 0.982] | | | |
| **DME** | 0.736 [0.629, 0.827] | 0.784 [0.684, 0.863] | 0.791 [0.689, 0.866] | | |
| **PEU** | 0.557 [0.336, 0.732] | 0.539 [0.336, 0.712] | 0.505 [0.268, 0.692] | 0.478 [0.263, 0.662] | |
| **PU** | 0.597 [0.369, 0.774] | 0.583 [0.345, 0.763] | 0.538 [0.282, 0.739] | 0.539 [0.307, 0.742] | 0.942 [0.890, 0.984] |

Note: DAG: Data Aggregation; DAN: Data Analysis; DIT: Data Interpretation; DME: Decision-Making Effectiveness; PEU: Perceived Ease of Use; PU- Perceived Usefulness

similarity, even though past studies have evidenced their conceptual distinction [7,25]. All HTMTs were significantly smaller from the value of 1 based on their 95% bias-corrected confidence intervals [43]. Hence, sufficient discriminant validity was supported.

## Measurement invariance assessment

The MICOM was performed to examine whether significant path differences between the two samples are attributed to substantive changes in the hypothesized relationships [35]. The MICOM results satisfied two out of three stages of assessment. First, configural invariance was established showing that both the Time 1 and Time 2 samples had identical model setup, data treatment, and algorithm settings of the model estimation. Second, the permutation tests substantiated that compositional invariance was established with identical composites (constructs) between Time 1 and Time 2 samples. Additionally, none of the composites had a correlation $c$ value significantly different from the value of 1 based on 5000 permutations at the two-tailed 0.05 significance level (see Table 4). Third, the means and variances equality assessment (see Tables 5 and 6) revealed that some of the composite's means and variances differed significantly between Time 1 and Time 2 samples. In conclusion, the presence of partial measurement invariance indicated that the between-sample comparison would be more meaningful than a pooled sample estimation. Therefore, MGA was justified for comparing changes in the structural path coefficients from Time 1 to Time 2.

## Structural model assessment

The structural model was assessed and satisfied all the evaluation criteria based on bootstrapped results of 5000 subsamples at the two-tailed 0.01 significance level. Then, the MGA was conducted to compare the hypothesized relationships between both periods. Please refer to the S3 Table for the alternative pooled sample estimation.

**Table 4. Composite invariance based on permutation test [step 2].**

| Construct | Correlation [c = 1] | 95% C.I. | p | Composite invariance? |
|---|---|---|---|---|
| **DAG** | 1.000 | [0.999, 1.000] | 0.478 | Yes |
| **DAN** | 1.000 | [0.999, 1.000] | 0.806 | Yes |
| **DIT** | 1.000 | [0.999, 1.000] | 0.945 | Yes |
| **DME** | 1.000 | [0.999, 1.000] | 0.268 | Yes |
| **PEU** | 1.000 | [0.999, 1.000] | 0.775 | Yes |
| **PU** | 1.000 | [0.995, 1.000] | 0.817 | Yes |

**Table 5. Equal mean assessment based on permutation test [step 3a].**

| Construct | Difference [D = 0] | 95% C.I. | p | Equal mean? |
|---|---|---|---|---|
| DAG | −0.496 | [-0.355, 0.351] | 0.005 | No |
| DAN | −0.673 | [-0.350, 0.347] | 0.000 | No |
| DIT | −0.634 | [-0.345, 0.351] | 0.000 | No |
| DME | −0.173 | [-0.356, 0.348] | 0.329 | Yes |
| PEU | −0.430 | [-0.341, 0.351] | 0.011 | No |
| PU | −0.471 | [-0.349, 0.350] | 0.006 | No |

**Table 6. Equal variance assessment based on permutation test [step 3b].**

| Construct | Difference [D = 0] | 95% C.I. | p | Equal variance? |
|---|---|---|---|---|
| DAG | 0.278 | [-0.710, 0.694] | 0.434 | Yes |
| DAN | −0.112 | [-0.627, 0.609] | 0.724 | Yes |
| DIT | −0.146 | [-0.712, 0.720] | 0.663 | Yes |
| DME | −0.144 | [-0.658, 0.664] | 0.682 | Yes |
| PEU | 0.486 | [-0.467, 0.463] | 0.039 | No |
| PU | 0.460 | [-0.604, 0.586] | 0.130 | Yes |

Table 7 presents the structural model results before (Time 1) and during the pandemic (Time 2). The hypothesized relationships were assessed based on size and significance using 95% bias-corrected and accelerated (BCa) bootstrap confidence intervals and $f2$ effect sizes. The effect sizes were interpreted according to Cohen's large (>0.35), moderate (>0.15), and weak (>0.02) $f2$ threshold [35].

No severe collinearity issue was found in the structural models at both periods, with all inner variance inflation factor (VIF) values less than 5 and mostly less than 3. However, probable collinearity was detected between DAN, DIT, and DME in Time 2, with inner VIF values close to 10 on hypotheses 4b and 4c. Please refer to the S4 Table for the inter-construct correlation matrix.

A possible remedy was to create a higher-order construct encompassing DAG, DAN, and DIT on DME [35], but the BA literature supports the conceptual distinction between these constructs. In this study, maintaining this conceptual distinction is crucial to compare the between-sample estimation based on a similar model structure and recommend ways of developing specific BA capabilities among ICU doctors. Furthermore, the Time 2 sample overlapped with the COVID-19 pandemic period, when ICU doctors were overworked and possibly struggled to distinguish between data analysis and interpretation for decision-making.

The PLS-SEM model fit was assessed based on SRMR, DULS, and DG with the 95% or 99% reference distribution. Overall, the models exhibited an acceptable fit with the values of SRMR smaller than 0.08 (Time 1: 0.072; Time 2: 0.057). All the discrepancy measures were below the 95% or 99% quantile of their reference distribution (Time 1: SRMR $0.072 < HI_{99}$ 0.077, $d_{ULS}$ $1.195 < HI_{99}$ 1.352, $d_G$ $1.347 < HI_{95}$ 1.620; Time 2: SRMR $0.057 < HI_{95}$ 0.057, $d_{ULS}$ $0.749 < HI_{95}$ 0.907, $d_G$ $1.267 < HI_{95}$ 1.762).

The PLS-SEM models had moderate predictive power to predict the outcome of DME in ICUs. The $R^2$ values indicated predictive performance that the PLS-SEM models explained 75.2% (Time 1) and 55.0% (Time 2) variation of DME. The predictive relevance of DME was higher in Time 1 ($Q^2$ 64.0%) than in Time 2 ($Q^2$ 39.7%). Based on the PLSpredict results, the PLS-SEM model outperformed the linear model by yielding smaller prediction errors based on root mean squared error (RMSE) for most of the indicators, except for two indicators in Time 1 and three indicators in Time 2.

## Comparing results before and during the pandemic

As shown in Table 7, PEU significantly improved doctors' BA capabilities for DAG, DAN, and DIT before the pandemic but not at all during the pandemic (H1a–H1c). While PU significantly enhanced DAG only before the pandemic (H2a), its influence shifted to DAN and DIT during the pandemic (H2b and H2c). The positive relationships between PEU and PU remained strong at both periods, with large effect sizes and no significant difference before and during the pandemic (H3). Among the three BA capabilities, DAG and DIT significantly influenced DME before the pandemic but lost their significance during the pandemic (H4a and H4c). DAN did not significantly influence DME before the pandemic, but later, DAN became the only capability significantly influencing DME with an increased effect size during the pandemic (H4b).

We found that PEU could not motivate doctors to use the EMR system for any BA capabilities development during the pandemic. Comparing the MGA results before and during the pandemic, it was inconclusive that the influence of PEU on DAG diminished by 29.8% (H1a). However, it was conclusive at the 5% significance level that the influence of PEU had significantly diminished by 55.1% on DAN (H1b) and 60.2% on DIT (H1c) through the pandemic.

We also found that PU became instrumental in motivating doctors to use the EMR system for DAN and DIT during the pandemic, with increased differences of 19.7% on DAN (H2b) and 30.5% on DIT (H2c). Although PU still positively influenced DAG during the pandemic, its influence diminished by 19.0% (H2a). However, none of the PU to BA changes were conclusive due to the insignificant between-sample differences before and during the pandemic.

Finally, concerning the outcome of this study, DAG improved DME by 11.4% during the pandemic (H4a), but this improvement was insignificant and inconclusive. In contrast, DAN significantly improved DME by 58.5% during the

**Table 7. Structural Model and MGA Results between Time 1 and Time 2 Samples.**

| | | Time 1 [n = 67] | | | Time 2 [n = 63] | | | Time 2 – Time 1 |
|---|---|---|---|---|---|---|---|---|
| Hypothesized Paths | | β | 95% BCa-CI | $f^2$ | β | 95% BCa-CI | $f^2$ | Δβ |
| H1a | PEU → DAG | 0.319 (0.086) * | [-0.065, 0.667] | 0.058 | 0.021 (0.923) | [-0.395, 0.441] | 0.000 | −0.298 (0.289) |
| H1b | PEU → DAN | 0.454 (0.007) *** | [0.086, 0.772] | 0.101 | −0.097 (0.646) | [-0.526, 0.316] | 0.004 | −0.551 (0.048) ** |
| H1c | PEU → DIT | 0.485 (0.009) *** | [0.123, 0.856] | 0.104 | −0.117 (0.519) | [-0.477, 0.228] | 0.005 | −0.602 (0.021) ** |
| H2a | PU → DAG | 0.418 (0.024) ** | [0.028, 0.773] | 0.099 | 0.228 (0.220) | [-0.200, 0.548] | 0.019 | −0.190 (0.479) |
| H2b | PU → DAN | 0.222 (0.258) | [-0.221, 0.580] | 0.024 | 0.419 (0.043) ** | [-0.068, 0.768] | 0.067 | 0.197 (0.480) |
| H2c | PU → DIT | 0.130 (0.599) | [-0.453, 0.522] | 0.007 | 0.435 (0.022) ** | [0.016, 0.767] | 0.072 | 0.305 (0.327) |
| H3 | PEU → PU | 0.847 (0.000) *** | [0.749, 0.909] | 2.543 | 0.815 (0.000) *** | [0.662, 0.893] | 1.977 | −0.032 (0.655) |
| H4a | DAG → DME | 0.181 (0.090) * | [-0.031, 0.388] | 0.050 | 0.295 (0.140) | [-0.093, 0.694] | 0.047 | 0.114 (0.607) |
| H4b | DAN → DME | 0.154 (0.299) | [-0.133, 0.438] | 0.024 | 0.739 (0.016) ** | [0.100, 1.306] | 0.116 | 0.585 (0.096) * |
| H4c | DIT → DME | 0.604 (0.000) *** | [0.363, 0.845] | 0.573 | −0.283 (0.288) | [-0.830, 0.236] | 0.018 | 0.887 (0.006) *** |
| | R² | 75.2% | | | 55.0% | | | |
| | Q² | 64.0% | | | 39.7% | | | |

Note: Significance level:

\*\*\* p < .01;

\*\* p < .05;

\* p < .10. PU: Perceived Usefulness. PEU: Perceived Ease of Use. DAG: Data Aggregation. DAN: Data Analysis. DIT: Data Interpretation. DME: Decision-Making Effectiveness.

β: standardized path coefficients.

95% BCa-CI: 95% Bias-Corrected and accelerated Bootstrap Confidence Interval.

$f^2$: Cohen's effect size.

Δβ: difference in standardized path coefficients from Time 1 to Time 2.

pandemic (H4b) and was conclusive at the 10% significance level. The decreasing effect of DIT on DME by 88.7% was also conclusive at the 1% significance level.

## Discussion

We found significant shifts in ICU doctors' perceptions of using the EMR system (i.e., PEU and PU) and their development of BA capabilities (i.e., DAG, DAN, and DIT) for decision-making effectiveness (DME) before and during the pandemic. We answered the two research questions based on our empirical results for the four main hypotheses.

Firstly, concerning doctors' technology perceptions, PEU diminished significantly and no longer influenced DAG, DAN, and DIT during the pandemic (Hypothesis 1). PU no longer influenced DAG but significantly influenced DAN and DIT during the pandemic (Hypothesis 2). PEU consistently motivated PU through the pandemic (Hypothesis 3), consistent with existing studies [16,44] and similar to the TAM postulation [25].

We infer that doctors had more time to learn and familiarize themselves with the EMR system before the pandemic. Although they perceived the EMR system was easy to use, they did not believe it was useful for developing the more complex BA capabilities, such as DAN and DIT. During the pandemic, doctors became inescapably dependent on the EMR system and realized its usefulness in coping with critical care priorities in an unprecedented situation. Higher PU means higher realization of the benefits of using the EMR system, such as increased accessibility to integrated healthcare data and reduced risk of human redundancy and errors [5,45]. Hence, doctors who believe the EMR system is easy to use will have a lower barrier to eventually use it for developing their BA capabilities [46].

Secondly, concerning doctors' development of BA capabilities, DAG positively contributed to DME while DIT negatively influenced DME during the pandemic. Neither relationship was significant and thus inconclusive. However, DAN was the only capability crucial for DME during the pandemic (Hypothesis 4).

We found that doctors shifted from DAG and DIT to relying on DAN only during the pandemic. This shift is consistent with recent studies where data analysis is essential to predict risk factors and understand prevalence trends [47,48]. Wang and Byrd [7] remarked that DAG is a precursor to using the EMR system functionalities for DAN and DIT. During the pandemic, when the ICU operated at overcapacity, the EMR system became indispensable in managing a massive influx of critically ill patients. Thus, doctors' ability to use the EMR system for data analysis becomes essential for making effective decisions.

With ICU doctors shifting their focus to DAN only for DME, much had changed in their approach to critically ill patients during the pandemic. In practice, specialized care was concentrated on patients suspected or confirmed with COVID-19 in a designated isolation room in the ICU [49]. ICU doctors were pressured to use the EMR system's data and functionalities within the quarantined area to quickly analyze each patient's case for an accurate diagnosis and treatment. Diagnostic accuracy requires doctors to analyze common symptoms and predict patients' survivability based on past medical history and current clinical records before considering specific individual needs to make an informed decision effectively. The effectiveness of decisions made for these patients requires the corroboration of input from other medical disciplines in multidisciplinary critical care management. The EMR system becomes a centralized communication hub between respective teams without piling up physical case notes in the ICU, thus reducing the risk of cross-infection. Minimizing time spent in the COVID-ICU also helps mitigate the risk of COVID-19 exposure among doctors.

Implementing the EMR system enables the digitization and management of large-scale medical and patient data. Specifically, harnessing the value of complex integrated healthcare data in an EMR system calls for one's digital and analytical skills when using the system [50]. As Price, Singer [51] showed, compared to paper-based systems, the EMR system removes redundancy by providing structured data management, such as health information, diagnostics, decision support, electronic communication between patients and doctors, and lightens administrative processes. These studies find that healthcare operational processes have benefited significantly from the digital integration and data analytics functionalities of an EMR system. The existing literature lacks evidence about the vital role of doctors' development of business analytics

capabilities in operating the EMR system [52]. This study bridges the gap between medical and information systems. Understanding doctors' perceptions of using an EMR system from the end users' perspective will better integrate the system functionalities with their operational and clinical practices for business value creation in hospitals [53,54].

Our findings offer several practical implications useful for healthcare providers. In the ICU setting, effective decision-making is a complex process requiring adequate knowledge and appropriate sources of information. Before the pandemic, doctors' perceived ease of using the EMR system contributed to developing their business analytics capabilities. Data aggregation and especially data interpretation assisted doctors in making effective healthcare decisions. However, during the pandemic, doctors emphasized the perceived usefulness of using the EMR system to develop their business analytics capabilities in data analysis and data interpretation. Our findings show that data analysis was particularly essential to decision-making effectiveness.

The EMR system serves as a reliable platform to use data in research that will be crucial and timely in addressing this novel coronavirus infection and in preparation for future pandemics. Accurate and complete patient records can be readily extracted from the centralized EMR database to conduct quick, evidence-based medical research to understand disease behavior. Furthermore, analyzing data from different medical disciplines can encourage multidisciplinary collaboration in holistically managing patient care. There requires a constant stream of evidence-based medical research that is contextually relevant and effective to improve every aspect of patient care in the public ICUs of Malaysia. Besides addressing the common onset of diseases, contextualization is vital to increase the overall survivability of ICU patients to accommodate specific local needs in producing patient-centered treatment plans.

Given the circumstances of critical care management, doctors' decision-making in the ICU requires more vigilance than in other hospital units. In this study, the value and benefit of using the EMR system are highly skewed toward critically ill patients who need a high level of personalized medical intervention against time. On the other hand, doctors in a regular ward could spend more time using the EMR system for data analysis to predict and prescribe disease-specific decisions. Therefore, we suggest hands-on training and engagement for ICU doctors to 'learn by doing' – practical use of the EMR system to develop their business analytics capabilities for data aggregation, analysis, and interpretation. These training initiatives will help doctors make effective decisions to predict patient-centric medical outcomes at an earlier stage of diagnosis and prescribe accurate treatment plans.

Information system adoption is often studied in the context of a profit-making industry, but less emphasis is placed on a nonprofit organization [55]. The cost of investing in the EMR system could deter nonprofit organizations, given the need to measure the return on investment [56]. Moreover, the incentive to adopt an information system differs since public tertiary hospitals have different objectives as nonprofit organizations. For example, a fast turnaround in discharging patients in the ICU can reduce operational costs and accommodate more patients. If the benefits of implementing an EMR system are not perceivable, achieving its widespread adoption will be strenuous unless its usage can effectively increase the organizational value [57].

Learning from the historical experience in the United States, the implementation of the EMR system in hospitals has initially struggled with "technology immaturity, health administrator focus on financial systems, application unfriendliness, and physician resistance" [58]. However, gradual acceptance of the EMR system occurred with a shift in doctors' perceptions, confidence in using the system, and improvement in computer skills [59]. Effective implementation of the EMR system requires responsible use and adaptation in the post-COVID era [47]. The pursuit of decision-making effectiveness can be achieved with a purposeful determination to view the EMR system as a business analytics-enabled architecture and strategically cultivate the use of the EMR system to develop doctors' business analytics capabilities.

## Conclusions, limitations, and future research

This study integrates the technology acceptance model [20] with the business analytics model in healthcare [7] as a single model. We specifically conceptualize and operationalize doctors' business analytics capabilities, thus contributing insights

to the users' perspective of using the EMR system for decision-making effectiveness in Malaysian public tertiary hospitals. When doctors find the EMR system easy to use, they consequently believe that using the EMR system is useful for developing their BA capabilities. Through the pandemic, doctors have less perceived the ease of using the EMR system but emphasize its usefulness for developing their data analysis and interpretation capabilities.

Our research design spans the pre-pandemic and pandemic periods, allowing us to compare the shifts in ICU doctors' technology perceptions and their increasing dependency on using the EMR system to develop business analytics capabilities for effective decision-making. The shift of perceptions occurred during the pandemic when ICU doctors took a step further to discover that the EMR system was useful in developing their data analysis capability for healthcare decisions. Data analysis is the only BA capability we found significantly influencing decision-making effectiveness in the ICU during the pandemic.

This study has some limitations. Finding the right time to access overwhelmed medical practitioners was a major challenge during the data collection. The country was in lockdown and medical practitioners were often overwhelmed. While the study aimed to administer a questionnaire survey to all medical practitioners working in the selected cluster, namely the ICUs, we could only reach out to medical practitioners who were involved in the immediate care of critically ill patients in the ICU and were present during our visit to the hospital. Our sampling frame was limited to medical practitioners in ICUs who are equipped with an EMR system in the Klang Valley region of Malaysia. Although all public tertiary hospitals have ICUs, not all hospitals have an established EMR system, especially in less developed parts of the region. Future research could consider extending the study to include a more comprehensive sample size encompassing respondents across the hospital departments and other ICUs with an established EMR system. Extended studies could also include nurses and hospital administrators.

## Supporting information

**S1 Table.  Appendix: Survey Instruments.**
(DOCX)

**S2 Table.  Appendix: Common Method Bias Assessment.**
(DOCX)

**S3 Table.  Appendix: Structural Model Results based on Pooled Sample.**
(DOCX)

**S4 Table.  Appendix: Inter-construct Correlation Matrix.**
(DOCX)

## Acknowledgments

We thank Dr. Melor Mansor (Head of Malaysian Anaesthesiology and Intensive Care Services and Head of Dept of Anaesthesiology and Intensive Care, Hospital Kuala Lumpur), Dr Shanti Rudra Deva (Consultant Intensivist and Head of Malaysian Adult Intensive Care Medicine Subspecialty Training, Hospital Kuala Lumpur), Dr. Mohd Rohisham bin Zainal Abidin (Head of Dept of Anaesthesiology and Intensive Care, Hospital Tengku Ampuan Rahimah Klang), Dr. Lee Chew Kiok (Consultant Intensivist, Hospital Sungai Buloh), and Dr. Premela Naidu a/p Sitaram (Head of Critical Care Services Unit, University Malaya Medical Centre) for allowing the study to be conducted in the respective Intensive Care Units.

## Author contributions

**Conceptualization:** Ewilly J. Y. Liew, Andrei O. J. Kwok.

**Data curation:** Ewilly J. Y. Liew, Sharon G. M. Koh, Yeh Han Poh, Shairil R. binti Ruslan, M. Shahnaz bin Hasan.

**Formal analysis:** Ewilly J. Y. Liew.

**Funding acquisition:** Ewilly J. Y. Liew.

**Investigation:** Ewilly J. Y. Liew, Andrei O. J. Kwok, Sharon G. M. Koh, Yeh Han Poh.

**Methodology:** Ewilly J. Y. Liew, Yeh Han Poh.

**Project administration:** Ewilly J. Y. Liew, Sharon G. M. Koh.

**Resources:** Ewilly J. Y. Liew, Shairil R. binti Ruslan, M. Shahnaz bin Hasan.

**Software:** Ewilly J. Y. Liew.

**Supervision:** Ewilly J. Y. Liew, Andrei O. J. Kwok.

**Validation:** Ewilly J. Y. Liew, Andrei O. J. Kwok, Sharon G. M. Koh, Yeh Han Poh.

**Writing – original draft:** Ewilly J. Y. Liew, Andrei O. J. Kwok, Sharon G. M. Koh, Yeh Han Poh.

**Writing – review & editing:** Ewilly J. Y. Liew, Andrei O. J. Kwok, Sharon G. M. Koh, Yeh Han Poh.

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
