## [Decision Letter · Decision Letter 0]

15 Feb 2023

PONE-D-22-18704Examining doctors’ business analytics capabilities in using the electronic medical record system for decision-making effectiveness in intensive care units: Impact of the COVID-19 pandemicPLOS ONE

Dear Dr. Liew,

Thank you for submitting your manuscript to PLOS ONE. After careful consideration, we feel that it has merit but does not fully meet PLOS ONE’s publication criteria as it currently stands. Therefore, we invite you to submit a revised version of the manuscript that addresses the points raised during the review process.

We look forward to receiving your revised manuscript.

Kind regards,

Humayun Kabir, MSc in Epidemiology

Academic Editor

PLOS ONE

Journal Requirements:

“This study is supported by the Global Asia in the 21st Century (GA21) Multidisciplinary Funding Platform (award number: GA-MA-16-L01) and COVID Kickstarter Fund Scheme (award number: Covid-2-2020) from Monash University Malaysia. The corresponding author, Ewilly, received both grant awards. The funders had no role in study design, data collection and analysis, decision to publish, or preparation of the manuscript.”

Reviewers' comments:

Reviewer's Responses to Questions

**Comments to the Author**

1. Is the manuscript technically sound, and do the data support the conclusions?

Reviewer #1: Partly

Reviewer #2: Yes

Reviewer #3: Yes

2. Has the statistical analysis been performed appropriately and rigorously? 

Reviewer #1: N/A

Reviewer #2: I Don't Know

Reviewer #3: Yes

3. Have the authors made all data underlying the findings in their manuscript fully available?

Reviewer #1: No

Reviewer #2: Yes

Reviewer #3: Yes

4. Is the manuscript presented in an intelligible fashion and written in standard English?

Reviewer #1: Yes

Reviewer #2: Yes

Reviewer #3: Yes

5. Review Comments to the Author

Reviewer #1: The abstract needs to be revised considering the context, challenge, aim, method, results and implications.

Research questions must be included.

More discussion is highly recommended for related works, research gaps, contributions and novelty.

More discussion is required for the results.

Research limitations and potential future directions need to be added.

More details must be clarified in the conclusion.

Reviewer #2: The use of an electronic medical record system in developing doctors' business analytics capabilities for more effective healthcare decisions in intensive care units prior to and during the COVID-19 pandemic is investigated in this manuscript. It is critical to address doctors' opinions of the EMR system's technology in terms of its ease of use and usefulness in connection to business analytics capabilities such as data aggregation, data analysis, and data interpretation. The authors analyze and evaluate the study questions thoroughly. They also highlight good theoretical and practical consequences.

Reviewer #3: This paper focuses on Doctors' business analytics capabilities in the use of the EMR system for decision making pre- and post-COVID-19. This is relevant in shaping good clinical practice. There are a few comments that need to be addressed:

1. The presentation creates the impression that this is a review paper. Since this is a Research paper, the literature review section is not needed. The literature provided should rather be used in the discussion section, which largely is not related to previous or other studies in its current format.

2. The Research model and hypothesis should be summarized. The copious hypothesis explanation is not necessary.

- Consider referencing figure 1 after healthcare setting (i.e. ".......in the health care setting (Figure 1)")

- The line from "The Research model is illustrated..." till the end of the section can be summarized as follows:

"We analyzed four (4) hypotheses:

1. PEU has a direct positive effect on using the EMR system for DAG, DAN, and DIT

2. PU ..................................................................

3. PEU..............................................................

4a. Using the EMR system for DAG has a direct positive effect on DME

4b. Using the EMR system for DAN .........................................

4c. Using the EMR system for DIT ......................................."

3. Methods:

- The sentence from "Comparing both periods......." till the end and Table 1 are best suited for the Results section.

4. Discussion

- As mentioned earlier, this section is largely not related to other or previous studies. Need revision.

5. Conclusions:

- Authors should summarize the conclusions into a key message related to Doctors' business analytics capabilities in using EMR system for decision making pre- and post-COVID-19.

6. Figure 1:

- In view of comment 2 above, there is no need for keeping H1a - H4c in the figure. The direction of the arrows is self-explanatory.

6. PLOS authors have the option to publish the peer review history of their article (what does this mean? ). If published, this will include your full peer review and any attached files.

**Do you want your identity to be public for this peer review?** For information about this choice, including consent withdrawal, please see our Privacy Policy .

Reviewer #1: No

Reviewer #2: No

Reviewer #3: **Yes: ** Abuaku, Benjamin

<quillbot-extension-portal></quillbot-extension-portal>

---

## [Author Response · Author response to Decision Letter 1]

8 Sep 2023

Please refer to the Response to Reviewers.docx for a properly formatted document.

Authors’ responses to reviewers’ comments:

Dear editors and reviewers,

First of all, thank you for providing us with the opportunity to submit a revised version of our manuscript, “Examining doctors’ business analytics capabilities in using the electronic medical record system for decision-making effectiveness in intensive care units: impact of the COVID-19 pandemic”, for publication consideration in PLOS ONE.

We thank all reviewers for their valuable comments. The positive feedback and compliments have certainly encouraged us to strive for excellence in our research. We very much appreciate the encouraging and useful comments given in response to our manuscript submission. Our revision takes all the suggested areas of improvement provided by the reviewers into account to improve our manuscript further. Below is a summary of our revision. Our detailed responses to each of the reviewers’ comments can be found on the following pages.

Please also find additional information requested by the academic editor in our cover letter. All relevant data are within the paper and available on the figshare public data repository. Please find the anonymous, private reviewer URL to our minimal data set here: https://figshare.com/s/d9cd0f8d81a4c284f919

We are confident that you will find we have been able to address all of the reviewers’ comments satisfactorily and hope that you will find the revised manuscript suitable for publication in your esteemed journal. Thank you again for inviting us to re-submit our manuscript.

Best regards,

*Anonymous authors*

 

Authors' response to Reviewer 1 Feedback

1. The abstract needs to be revised considering the context, challenge, aim, method, results and implications.

We thank you for the positive feedback and compliments that have certainly encouraged us to strive for excellence in our research. We very much appreciate your encouraging and useful comments given in response to our manuscript submission. Our revision considers all your suggested areas of improvement to improve our manuscript further.

Thank you for your comment. We have revised the abstract thoroughly as suggested. In particular, we incorporated and/or enhanced the study’s challenge (lines 3-4), aim (lines 4-6), research questions (lines 7-9), context (lines 10-12), methods (lines 12-14), results (lines 14-19), and implications (lines 19-24) in the abstract.

2. Research questions must be included.

We have included the research questions in the abstract (p.1 lines 7-9). We have also indented and italicized the research questions for better clarity in the introduction section (p.3 lines 57-60).

3. More discussion is highly recommended for related works, research gaps, contributions and novelty.

More discussion is required for the results.

Thank you for your recommendation. We have thoroughly extended the discussion by linking to related works, research gaps, contributions, and novelty. Please refer to the revised Discussion section on pp.22-29 in the Manuscript.

4. Research limitations and potential future directions need to be added.

We renamed the conclusions section to “Conclusions, limitations, and future research” to better reflect the content in this section. We further clarified the research limitations and potential future directions related to our study (pp.30-31, lines 568-580).

5. More details must be clarified in the conclusion.

We added a key message summarizing the study’s conclusion (pp.29-30, lines 543-548):

When doctors find the EMR system easy to use, they consequently believe that using the EMR system is useful for developing their data-related capabilities. Through the pandemic, doctors perceive less the ease of using the EMR system but emphasize its usefulness for developing their data analysis and interpretation capabilities. More specifically, doctors’ data analysis capability becomes the sole driver for making effective decisions in ICU during the pandemic.

We further clarified the research limitations and potential future directions related to our study (pp.30-31, lines 568-580).

 

Authors' response to Reviewer 2 Feedback

The use of an electronic medical record system in developing doctors' business analytics capabilities for more effective healthcare decisions in intensive care units prior to and during the COVID-19 pandemic is investigated in this manuscript. It is critical to address doctors' opinions of the EMR system's technology in terms of its ease of use and usefulness in connection to business analytics capabilities such as data aggregation, data analysis, and data interpretation. The authors analyze and evaluate the study questions thoroughly. They also highlight good theoretical and practical consequences.

We thank you for the positive feedback and compliments that have certainly encouraged us to strive for excellence in our research. We very much appreciate the reviewer’s appreciation of our work. Our revision considers all the suggested areas of improvement provided by the reviewers to improve our manuscript further.

 

Authors' response to Reviewer 3 Feedback

This paper focuses on Doctors' business analytics capabilities in the use of the EMR system for decision making pre- and post-COVID-19. This is relevant in shaping good clinical practice. There are a few comments that need to be addressed:

1. The presentation creates the impression that this is a review paper. Since this is a Research paper, the literature review section is not needed. The literature provided should rather be used in the discussion section, which largely is not related to previous or other studies in its current format.

We thank you for the positive feedback and compliments that have certainly encouraged us to strive for excellence in our research. We very much appreciate your encouraging and useful comments regarding our work. Our revision considers all your suggested areas of improvement to improve our manuscript further.

We have removed the literature review as suggested and extended the discussion section by carefully relating to previous studies (pp.22-29 in the Manuscript).

We are confident that the revised manuscript now creates a more appropriate impression as a Research paper.

2. The Research model and hypothesis should be summarized. The copious hypothesis explanation is not necessary.

- Consider referencing figure 1 after healthcare setting (i.e. ".......in the health care setting (Figure 1)")

- The line from "The Research model is illustrated..." till the end of the section can be summarized as follows:

"We analyzed four (4) hypotheses:

1. PEU has a direct positive effect on using the EMR system for DAG, DAN, and DIT

2. PU ..................................................................

3. PEU..............................................................

4a. Using the EMR system for DAG has a direct positive effect on DME

4b. Using the EMR system for DAN .........................................

4c. Using the EMR system for DIT ......................................."

We have summarized as suggested.

We referenced (Fig 1) after “the healthcare setting” and removed the following line (p.5 in the Manuscript).

We added “we analysed four (4) main hypotheses” following Fig 1 (p.5 lines 106-117).

H1: Perceived ease of use (PEU) has a direct positive effect on using the EMR system for data aggregation (DAG; H1a), data analysis (DAN; H1b), and data interpretation (DIT; H1c).

H2: Perceived usefulness (PU) has a direct positive effect on using the EMR system for data aggregation (DAG; H2a), data analysis (DAN; H2b), and data interpretation (DIT; H2c).

H3: Perceived ease of use (PEU) has a direct positive effect on the perceived usefulness (PU) of using the EMR system.

H4: Using the EMR system for data aggregation (DAG; H4a), data analysis (DAN; H4b), and data interpretation (DIT; H4c) has a direct positive effect on decision-making effectiveness (DME).

While we now have four main hypotheses (H1-H4), we also retained the notations for sub-hypotheses (H1a, H1b,…etc) in Fig 1 and the Manuscript. The inclusion would ensure clarity for readers when cross-checking between Fig 1 and the hypothesized paths tested in the structural model (especially Table 7, pp.18-19) and reported under the structural model assessment section (pp. 17-21). The notations for sub-hypotheses would also help readers to follow through the discussion section (pp. 22-29) as we discussed the interplay between the sub-dimensions of technology perceptions (PEU, PU) and business analytics capabilities (DAG, DAN, DIT) on influencing decision-making effectiveness (DME). We hope the reviewer will find our explanation satisfactory to retain the notation for sub-hypotheses in the Manuscript.

3. Methods:

- The sentence from "Comparing both periods......." till the end and Table 1 are best suited for the Results section.

Thank you for pointing this out. We have revised as suggested. Table 1 and its corresponding text have been moved from the Methods section to the Results section (pp.9-11, lines 215-229).

4. Discussion

- As mentioned earlier, this section is largely not related to other or previous studies. Need revision.

Thank you for your comment. We have removed the literature review as suggested and extended the discussion section by carefully relating to previous studies. Please refer to the revised Discussion section on pp.22-29 in the Manuscript.

5. Conclusions:

- Authors should summarize the conclusions into a key message related to Doctors' business analytics capabilities in using EMR system for decision making pre- and post-COVID-19.

Thank you for your useful suggestion. We added a key message summarizing the study’s conclusion (pp.29-30, lines 543-548):

When doctors find the EMR system easy to use, they consequently believe that using the EMR system is useful for developing their data-related capabilities. Through the pandemic, doctors perceive less the ease of using the EMR system but emphasize its usefulness for developing their data analysis and interpretation capabilities. More specifically, doctors’ data analysis capability becomes the sole driver for making effective decisions in ICU during the pandemic.

6. Figure 1:

- In view of comment 2 above, there is no need for keeping H1a - H4c in the figure. The direction of the arrows is self-explanatory.

Thank you for your suggestion. We respectfully decided not to revise Fig 1.

Although we have listed the four main hypotheses as H1-H4 following Fig 1 (p.5 lines 106-117), we decided to retain the notations for sub-hypotheses (H1a, H1b,…etc) in Fig 1 and the Manuscript.

While the arrow direction in Fig 1 is self-explanatory, keeping the notation for sub-hypotheses (H1a – H4c) would ensure clarity for readers when cross-checking between Fig 1 and the hypothesized paths tested in the structural model (especially Table 7, pp.18-19) and reported under the structural model assessment section (pp. 17-21).

The notations for sub-hypotheses would also help readers to follow through the discussion section (pp.22-29) as we discussed the interplay between the sub-dimensions of technology perceptions (PEU, PU) and business analytics capabilities (DAG, DAN, DIT) on influencing decision-making effectiveness (DME). We hope the reviewer will find our explanation satisfactory to retain the notation for sub-hypotheses in the Manuscript.

--- End of Authors’ responses to reviewers’ comment ---

---

## [Decision Letter · Decision Letter 1]

19 Nov 2023

PONE-D-22-18704R1Examining doctors’ business analytics capabilities in using the electronic medical record system for decision-making effectiveness in intensive care units: Impact of the COVID-19 pandemicPLOS ONE

Dear Dr. Liew,

Thank you for submitting your manuscript to PLOS ONE. After careful consideration, we feel that it has merit but does not fully meet PLOS ONE’s publication criteria as it currently stands. Therefore, we invite you to submit a revised version of the manuscript that addresses the points raised during the review process.

We look forward to receiving your revised manuscript.

Kind regards,

Humayun Kabir

Academic Editor

PLOS ONE

Reviewers' comments:

Reviewer's Responses to Questions

**Comments to the Author**

1. If the authors have adequately addressed your comments raised in a previous round of review and you feel that this manuscript is now acceptable for publication, you may indicate that here to bypass the “Comments to the Author” section, enter your conflict of interest statement in the “Confidential to Editor” section, and submit your "Accept" recommendation.

Reviewer #3: All comments have been addressed

Reviewer #4: (No Response)

2. Is the manuscript technically sound, and do the data support the conclusions?

Reviewer #3: (No Response)

Reviewer #4: Partly

3. Has the statistical analysis been performed appropriately and rigorously? 

Reviewer #3: (No Response)

Reviewer #4: I Don't Know

4. Have the authors made all data underlying the findings in their manuscript fully available?

Reviewer #3: (No Response)

Reviewer #4: (No Response)

5. Is the manuscript presented in an intelligible fashion and written in standard English?

Reviewer #3: (No Response)

Reviewer #4: No

6. Review Comments to the Author

Reviewer #3: (No Response)

Reviewer #4: Thank for your the opportunity to review this manuscript. The topic is interesting and important to clinicians and the study itself seems to have been rigorously done. However, if the target audience includes clinicians, I worry that in its current format, the manuscript is difficult to follow and excessively lengthy.

As a clinician and researcher, I found this paper somewhat difficult to follow, overly wordy in many places with words and phrases. I do acknowledge that the edits in response to the previous reviewers have improved the manuscript.

I think the manuscript would benefit from copy editing. I will try to provide some specific suggestions but cannot address all of the statements/ sections that need more brevity.

Some suggestions for the authors to consider:

General Comments

- As a starting point, perhaps the authors can consistently use active voice rather than complex passive sentences.

- Also, the structure of the sections are not standard for medical literature. It's more akin to a graduate thesis. Can the authors re-evaluate and re-structure the manuscript to be more familiar to medical readers?

- The abstract is not structured with subheadings. Is this not a requirement by the journal?

- Would suggest using more simple statements without added adverbs. For example, the discussion starts with “the empirical results evidence…” These words are all saying the same thing. How about simply, “we found that…” Another example is in the Discussion (1st para): "... significant predictors of doctors' behavioral intension to use the system..." What are behavioral intentions? There are numerous other examples of this issue and I hope the authors can review the manuscript and address these.

Introduction

- If the authors' objective was to evaluate the perceptions of doctors on the usability… etc of this EMR, could the authors consider a more succinct Introduction that simply ends with this objective statement?

- The final sections of the Introduction describes study methodology. Can the authors just include this in the methods section so the Intro ends with study objectives?

- Business analytics capabilities is a prominent feature of the study but is not adequately defined. Please clearly define this in the Introduction and indicate its relationship with EMR usage. This is currently highly unclear. To provide context for readers, are there any examples of the relevance of this in critical care or even in clinical medicine in general?

Methods

- Please describe the difference between specialists, medical officers, and house officers as these terms are used in some countries but not others.

- Please describe what common methods bias is, if this article is intended for a medical audience, this is likely not familiar to them.

Results

- The results section dives right into the main findings without describing the sample, response rates, etc. Please describe the study sample first (currently somewhat described in second paragraph).

- What were the response rates? (ie, how many were invited and how many responded partially and fully?)

Discussion

- The Discussion covers the literature well, but is overly lengthy as well.

- Is there a need for the Discussion to include a page on "Theoretical implications and contributions" and "Practical implications and contributions"?

7. PLOS authors have the option to publish the peer review history of their article (what does this mean? ). If published, this will include your full peer review and any attached files.

**Do you want your identity to be public for this peer review?** For information about this choice, including consent withdrawal, please see our Privacy Policy .

Reviewer #3: No

Reviewer #4: No

---

## [Author Response · Author response to Decision Letter 2]

19 May 2024

Please also refer to the Response to Reviewers R2.docx document.

PONE-D-22-18704

Examining doctors’ business analytics capabilities in using the electronic medical record system for decision-making effectiveness in intensive care units: impact of the COVID-19 pandemic

Authors’ responses to reviewers’ comments:

Dear editors and reviewers,

First of all, thank you for providing us with the opportunity to submit a second revised version of our manuscript, “Examining doctors’ business analytics capabilities in using the electronic medical record system for decision-making effectiveness in intensive care units: impact of the COVID-19 pandemic”, for publication consideration in PLOS ONE.

We thank all reviewers for their valuable comments. The positive feedback and compliments have certainly encouraged us to strive for excellence in our research. We very much appreciate the encouraging and useful comments regarding our manuscript submission. Our revision takes all the suggested areas of improvement the reviewers provided to improve our manuscript further. Below is a summary of our revision. Our detailed responses to each of the reviewers’ comments can be found on the following pages.

In this second revision, thanks to the 4th reviewer, we enhanced the readability of our paper significantly, making it suitable for medical readers and the general audience. We have copyedited this second revised manuscript to (i) consistently use active voice where possible, except when reporting methods and results, (ii) simplify complex passive sentences with coherent and concise wordings, and (iii) re-structure/re-organize some parts of the manuscript to be more familiar to medical readers. We trust this copyedited manuscript now reads well, is coherent, and is easier to follow.

Please be assured that we have also carefully preserved the improvements we have previously made in our first revised manuscript. The additional information requested by the academic editor remains as communicated in our cover letter. All relevant data are included within the paper and available in the figshare public data repository. Please find the anonymous, private reviewer URL to our minimal data set here: https://figshare.com/s/d9cd0f8d81a4c284f919

We are confident that you will find we have been able to address all of the reviewers’ comments satisfactorily and hope that you will find the second revised manuscript suitable for publication in your esteemed journal. Thank you again for inviting us to re-submit our manuscript.

Best regards,

*Anonymous authors*

Reviewer 3 Feedback

All comments have been addressed. No further response to the review questions #2 to #6.

Authors' responses:

We thank you for the assurance and the compliments that we have addressed all the comments in your previous review, and no further revision is required.

Reviewer 4 Feedback

Thank you for the opportunity to review this manuscript. The topic is interesting and important to clinicians and the study itself seems to have been rigorously done. However, if the target audience includes clinicians, I worry that in its current format, the manuscript is difficult to follow and excessively lengthy.

As a clinician and researcher, I found this paper somewhat difficult to follow, overly wordy in many places with words and phrases. I do acknowledge that the edits in response to the previous reviewers have improved the manuscript.

I think the manuscript would benefit from copy editing. I will try to provide some specific suggestions but cannot address all of the statements/ sections that need more brevity.

Authors' responses:

Thank you for your comments. We are encouraged by your positive feedback that our topic is interesting and important to clinicians like your good self. We very much appreciate your useful suggestions as a clinician and researcher. Our revision considers all your suggested areas of improvement to improve our manuscript further.

We have copyedited our manuscript to enhance its structure and readability, making it suitable for medical readers and the general audience. You may find more specific changes we have made to this revised manuscript in our following responses to your comments.

Please be assured that while copyediting this second revised manuscript, we have also carefully preserved the improvements we made in our first revised manuscript.

General Comments

- As a starting point, perhaps the authors can consistently use active voice rather than complex passive sentences.

- Also, the structure of the sections are not standard for medical literature. It's more akin to a graduate thesis. Can the authors re-evaluate and re-structure the manuscript to be more familiar to medical readers?

- Would suggest using more simple statements without added adverbs. For example, the discussion starts with “the empirical results evidence…” These words are all saying the same thing. How about simply, “we found that…” Another example is in the Discussion (1st para): "... significant predictors of doctors' behavioral intension to use the system..." What are behavioral intentions? There are numerous other examples of this issue and I hope the authors can review the manuscript and address these.

Authors' responses:

We have copyedited this second revised manuscript to (i) consistently use active voice and first-person point-of-view (“we…”) where possible, except when reporting methods and results, (ii) simplify complex passive sentences with coherent and concise wordings, and (iii) re-structure/re-organize some parts of the manuscript to be more familiar to medical readers. We carefully reviewed and revised the manuscript to ensure consistent usage of words, format, and structure to enhance readability.

We also removed unnecessary words that might cause confusion to our readers. For example, we refrained from using ‘behavioral intention’, an outcome variable widely found in the information system literature but not an outcome our study measures.

We trust this copyedited manuscript now reads well, is coherent, and is easier to follow.

Abstract

- The abstract is not structured with subheadings. Is this not a requirement by the journal?

Thank you for your comment. We have revised the abstract according to your suggestions.

Authors' responses:

Previously, our first revised abstract was written following a standard structure outlining the study’s challenge, aim, research questions, context, methods, results, and implications.

Now, we added the subheadings to guide readers through our abstract.

We referred to samples of PLoS ONE published articles for the abstract and manuscript format, e.g., Vaithilingam et al. (2023) at https://doi.org/10.1371/journal.pone.0282520

Introduction

- If the authors' objective was to evaluate the perceptions of doctors on the usability… etc of this EMR, could the authors consider a more succinct Introduction that simply ends with this objective statement?

- The final sections of the Introduction describes study methodology. Can the authors just include this in the methods section so the Intro ends with study objectives?

- Business analytics capabilities is a prominent feature of the study but is not adequately defined. Please clearly define this in the Introduction and indicate its relationship with EMR usage. This is currently highly unclear. To provide context for readers, are there any examples of the relevance of this in critical care or even in clinical medicine in general?

Authors' responses:

Thank you for your suggestions. We have restructured and reworked the Introduction to enhance the study’s focus and relevance to medical readers. In particular, we restructured the logical flow of the Introduction to align with our study’s objective—evaluating doctors’ perception of using the EMR system to develop business analytics (BA) capabilities for effective decision-making.

We ended the Introduction with a strong objective statement and two research questions (recommended by the previous reviewers). And we moved the final sections of the Introduction to more relevant sections of the manuscript i.e., the Conceptual model and the Methods sections.

The Introduction has an improved structure, briefly described as follows.

We started our introduction by emphasizing the advancement and purpose of using an EMR system in hospitals as a vital decision support tool. We incorporated the context of public tertiary hospitals that serve non-profit-driven business objectives to define and clarify decision-making effectiveness (DME) as the outcome variable of our study.

Then, we lent support from existing information systems (IS), health IS, and business analytics (BA) literature to understand why the use of EMR systems can relate with BA capabilities. We logically unfolded why we should examine the BA capabilities development from the end users’ perspectives, i.e., individual doctors behind using an EMR system.

Finally, we emphasized our study’s objective to address the arising challenge/problem we hoped to examine---when doctors are ambiguous about the benefits of using the EMR system and lack the BA-related skill sets to derive these benefits from using the EMR system for healthcare decisions.

Methods

- Please describe the difference between specialists, medical officers, and house officers as these terms are used in some countries but not others.

- Please describe what common methods bias is, if this article is intended for a medical audience, this is likely not familiar to them. Thank you for your suggestions. W

Authors' responses:

After evaluating our study’s objective, we decided to remove this potentially confusing clause of (specialists, medical officers, and house officers) when describing our data collection process. Instead we clarified by adding the following text (2nd paragraph under data collection):

We employed the cluster random sampling method to survey ICU doctors currently attending to critical care in a public tertiary hospital and using the EMR system for decision-making. Doctors refer to individuals with MD degrees and are the primary decision-makers about patient care and diagnosis (Skyes & Venkatesh, 2011).

We simplified the statistical analysis subheading in two ways. Firstly, the procedures for performing the PLS-SEM analysis are integrated directly into the results in the Result section where appropriate. Secondly, the procedures for assessing common method bias are moved to the supporting information in the S2 Table. We also added a line to describe what and why the common method bias assessment is necessary to ensure robustness in our analysis.

Results

- The results section dives right into the main findings without describing the sample, response rates, etc. Please describe the study sample first (currently somewhat described in second paragraph).

- What were the response rates? (ie, how many were invited and how many responded partially and fully?) Thank you for pointing out a potential improvement to our manuscript.

Authors' responses:

We revised the first two paragraphs into a single paragraph under the Results section to present summary statistics of the final sample consisting of 130 valid responses.

For clarification, we have previously included the samples and sampling procedure under the Methods > Data collection and sampling design. With the revision of the first paragraph in the Result, we mitigate the risk of readers missing the study’s sample description.

As described in the manuscript (2nd paragraph under data collection), the survey invitation was sent out to all doctors involved in critical care at the selected ICUs. Time 1 survey was distributed using paper-based and online questionnaires before the pandemic. The time 2 survey was administered using online questionnaires only during the pandemic due to access limitations.

While the study aimed to administer a questionnaire survey to all doctors working in the selected cluster, namely the ICUs, we could only reach out to doctors involved in the immediate care of critically ill patients in the ICU who were present during our visit to the hospital. Our sampling frame was also limited to doctors in ICUs equipped with an EMR system in the Klang Valley region of Malaysia. Regrettably, we do not have the base value to compute our response rate because the full list of ICU doctors on duty was unavailable, especially due to the high staff movement during the pandemic.

As such, we clarify the responses by adding the following text (2nd paragraph under data collection): We could only recruit ICU doctors who were present during our visit to the hospital before the pandemic (Time 1) and through snowballing contacts among the closely knitted ICU doctor groups in the hospital during the pandemic (Time 2).

Discussion

- The Discussion covers the literature well, but is overly lengthy as well.

- Is there a need for the Discussion to include a page on "Theoretical implications and contributions" and "Practical implications and contributions"? Thank you for your recommendation. The Discussion section was thoroughly extended in our first revision, linking it with related works, research gaps, contributions, and novelty as suggested by previous reviewers. Unwittingly, this section became overly lengthy.

Authors' responses:

In this second revision, we put considerable effort into re-evaluating and restructuring the Discussion section. Where appropriate, we moved some discussions to the Introduction and the Conceptual model sections so that readers could follow the study’s research gaps and novelty early in the manuscript as we unfolded the examination. We trimmed down unnecessary words and section subheadings that might cause confusion to readers. We also simplified sentences to craft a coherent and logical Discussion section.

--- End of Authors’ responses to reviewers’ comment ---

---

## [Decision Letter · Decision Letter 2]

5 Nov 2024

PONE-D-22-18704R2Examining doctors’ business analytics capabilities in using the electronic medical record system for decision-making effectiveness in intensive care units: Impact of the COVID-19 pandemicPLOS ONE

Dear Dr. Liew,

Thank you for submitting your manuscript to PLOS ONE. After careful consideration, we feel that it has merit but does not fully meet PLOS ONE’s publication criteria as it currently stands. Therefore, we invite you to submit a revised version of the manuscript that addresses the points raised during the review process.

**Only one of the reviewers provided additional feedback, specifically suggesting a few minor grammatical adjustments to enhance readability. Please find the reviewer’s comments below.**

We look forward to receiving your revised manuscript.

Kind regards,

Jose Palma, Ph.D.

Academic Editor

PLOS ONE

Journal Requirements:

Reviewers' comments:

Reviewer's Responses to Questions

**Comments to the Author**

1. If the authors have adequately addressed your comments raised in a previous round of review and you feel that this manuscript is now acceptable for publication, you may indicate that here to bypass the “Comments to the Author” section, enter your conflict of interest statement in the “Confidential to Editor” section, and submit your "Accept" recommendation.

Reviewer #4: All comments have been addressed

2. Is the manuscript technically sound, and do the data support the conclusions?

Reviewer #4: Yes

3. Has the statistical analysis been performed appropriately and rigorously? 

Reviewer #4: I Don't Know

4. Have the authors made all data underlying the findings in their manuscript fully available?

Reviewer #4: Yes

5. Is the manuscript presented in an intelligible fashion and written in standard English?

Reviewer #4: Yes

6. Review Comments to the Author

**Reviewer #4:**  Thank you for the opportunity to review this revised manuscript. I appreciate that the authors have addressed all of my comments in a comprehensive manner. I think the manuscript is easier to understand for those with less content expertise and more appropriate for a general medical audience.

Only a few minor grammatical suggestions below, but otherwise, I think the manuscript is much improved.

Introduction

Page 3 - "EMR system becomes a vital decision support tool" seem like an incomplete sentence after the revisions.

Also Page 3 - "More so in the ICU setting, where mortality and morbidity risks are higher than in other wards of a hospital." This is an incomplete sentence.

Methods

Page 16 - First para - "We design the questionnaire..." should be "We designed..."

7. PLOS authors have the option to publish the peer review history of their article (what does this mean? ). If published, this will include your full peer review and any attached files.

**Do you want your identity to be public for this peer review?** For information about this choice, including consent withdrawal, please see our Privacy Policy .

Reviewer #4: No

---

## [Author Response · Author response to Decision Letter 3]

20 Dec 2024

PONE-D-22-18704

Examining doctors’ business analytics capabilities in using the electronic medical record system for decision-making effectiveness in intensive care units: impact of the COVID-19 pandemic

Authors’ responses to reviewers’ comments:

Dear editors and reviewers,

First of all, thank you for providing us with the opportunity to submit a third revised version of our manuscript, “Examining doctors’ business analytics capabilities in using the electronic medical record system for decision-making effectiveness in intensive care units: impact of the COVID-19 pandemic”, for publication consideration in PLOS ONE.

We thank the 4th reviewer for accepting our revision in the previous revised manuscript. We are very much encouraged by the kind and useful comments. In this version, our revision takes all the suggestions provided by the 4th reviewer to improve our manuscript to a publication-ready version. Please find a summary of our revision on the following pages.

We are confident that you will find we have been able to address all of the reviewer’s comments satisfactorily and hope that you will find the third revised manuscript ready for publication in your esteemed journal. Thank you again for inviting us to re-submit our manuscript.

Best regards,

*Anonymous authors*

Reviewers' comments:

Reviewer's Responses to Questions

Comments to the Author

1. If the authors have adequately addressed your comments raised in a previous round of review and you feel that this manuscript is now acceptable for publication, you may indicate that here to bypass the “Comments to the Author” section, enter your conflict of interest statement in the “Confidential to Editor” section, and submit your "Accept" recommendation.

Reviewer #4: All comments have been addressed

2. Is the manuscript technically sound, and do the data support the conclusions?

Reviewer #4: Yes

3. Has the statistical analysis been performed appropriately and rigorously?

Reviewer #4: I Don't Know

4. Have the authors made all data underlying the findings in their manuscript fully available?

Reviewer #4: Yes

5. Is the manuscript presented in an intelligible fashion and written in standard English?

Reviewer #4: Yes

6. Review Comments to the Author

Reviewer #4: Thank you for the opportunity to review this revised manuscript. I appreciate that the authors have addressed all of my comments in a comprehensive manner. I think the manuscript is easier to understand for those with less content expertise and more appropriate for a general medical audience.

Only a few minor grammatical suggestions below, but otherwise, I think the manuscript is much improved.

Introduction

Page 3 - "EMR system becomes a vital decision support tool" seem like an incomplete sentence after the revisions.

Also Page 3 - "More so in the ICU setting, where mortality and morbidity risks are higher than in other wards of a hospital." This is an incomplete sentence.

Methods

Page 16 - First para - "We design the questionnaire..." should be "We designed..."

7. PLOS authors have the option to publish the peer review history of their article (what does this mean?). If published, this will include your full peer review and any attached files.

Do you want your identity to be public for this peer review? For information about this choice, including consent withdrawal, please see our Privacy Policy.

Reviewer #4: No

--------------

Reviewer 4 Comment:

Thank you for the opportunity to review this revised manuscript. I appreciate that the authors have addressed all of my comments in a comprehensive manner. I think the manuscript is easier to understand for those with less content expertise and more appropriate for a general medical audience.

Only a few minor grammatical suggestions below, but otherwise, I think the manuscript is much improved.

Authors' Response:

Thank you for your comments. We very much appreciate your encouragement that the manuscript is now easier to understand for a general medical audience. Our revision considers all your suggested areas of improvement to improve our manuscript toward a publication-ready version.

We have copyedited our manuscript to enhance its structure and readability, addressing all the grammatical suggestions.

We also corrected the sequencing of our in-text citations to a correct order. You may find our amendments to this revised manuscript in our following responses to your comments.

Please be assured that while copyediting this third revised manuscript, we have also carefully preserved the improvements we made in our first revised manuscript.

Introduction

Page 3 - "EMR system becomes a vital decision support tool" seem like an incomplete sentence after the revisions.

Also Page 3 - "More so in the ICU setting, where mortality and morbidity risks are higher than in other wards of a hospital." This is an incomplete sentence. Thank you for your suggestion. We have revised the lines to

“In the care and management of critically ill patients, the EMR system becomes a vital support tool aiding doctors in treatment plans”.

“More so in the setting of the intensive care unit (ICU), where critically ill are cared for, decisions that affect mortality and morbidity outcomes are higher compared to the general wards which house relatively stable patients.”

--------------

Reviewer's comment:

Methods

Page 16 - First para - "We design the questionnaire..." should be "We designed..."

Authors' Response:

Thank you for pointing out, we have corrected the grammatical error.

--- End of Authors’ responses to reviewers’ comment ---

---

## [Editor Report · Decision Letter 3]

7 Jan 2025

Examining doctors’ business analytics capabilities in using the electronic medical record system for decision-making effectiveness in intensive care units: Impact of the COVID-19 pandemic

PONE-D-22-18704R3

Dear Dr. Liew,

We’re pleased to inform you that your manuscript has been judged scientifically suitable for publication and will be formally accepted for publication once it meets all outstanding technical requirements.

Kind regards,

Jose Palma, Ph.D.

Academic Editor

PLOS ONE
---

## [Editor Report · Acceptance letter]

PONE-D-22-18704R3

PLOS ONE

Dear Dr. Liew,

I'm pleased to inform you that your manuscript has been deemed suitable for publication in PLOS ONE. Congratulations! Your manuscript is now being handed over to our production team.

Kind regards,

on behalf of

Mr. Jose Palma

Academic Editor

PLOS ONE